ecology, evolution, genomics

symbiosis, population genetics, local adaptation

**Author for correspondence:**
Brendan H. Cornwell
e-mail: bhcornwell@ucdavis.edu

†Present address: Hopkins Marine Station of Stanford University, Pacific Grove, CA 93950, USA

# Genetic structure in the endosymbiont *Breviolum 'muscatinei'* is correlated with geographical location, environment and host species

Brendan H. Cornwell[1],[†] and Luis Hernández[2]

[1]Center for Population Biology, University of California, Davis, CA 95616, USA
[2]Departamento de Ciencias Marinas y Costeras, Universidad Autónoma de Baja California Sur, La Paz, Baja California Sur, Mexico

BHC, 0000-0001-7839-8379

Corals and cnidarians form symbioses with dinoflagellates across a wide range of habitats from the tropics to temperate zones. Notably, these partnerships create the foundation of coral reef ecosystems and are at risk of breaking down due to climate change. This symbiosis couples the fitness of the partners, where adaptations in one species can benefit the holobiont. However, the scales over which each partner can match their current—and future—environment are largely unknown. We investigated population genetic patterns of temperate anemones (*Anthopleura* spp.) and their endosymbiont *Breviolum 'muscatinei'*, across an extensive geographical range to identify the spatial scales over which local adaptation is possible. Similar to previously published results, two solitary host species exhibited isolation by distance across hundreds of kilometres. However, symbionts exhibited genetic structure across multiple spatial scales, from geographical location to depth in the intertidal zone, and host species, suggesting that symbiont populations are more likely than their hosts to adaptively mitigate the impact of increasing temperatures.

## 1. Introduction

There is mounting evidence that associations with microbial symbiont partners can dramatically expand the niche space of a host species [1–3]. Especially in situations where host populations cannot locally adapt (e.g. due to gene flow causing an influx of maladaptive alleles), associating with a symbiont specializing on a specific environment can increase holobiont performance, and possibly the fitness of the host [4].

Together, hosts and symbiont genotypes interact across a landscape that varies in space and time, or a 'geographical mosaic' [5]. Local adaptation, coevolution or both, can increase the fitness of either partner across the geographical mosaic [6,7]. These forces can cause varying partner quality within and between populations, which can affect the stability of the partnership in novel environmental conditions [8]. However, the ecologically relevant spatial scales that influence partner quality are often unclear, as 'local' environments for symbionts range from different locations within a host organism [9] to hundreds or thousands of kilometers [6,10]. Defining the scales over which partner quality changes due to differences in the environment across the geographical mosaic is crucial; for example, partnerships such as the Symbiodiniaceae—cnidarian endosymbiosis, and the coral reef ecosystem it supports, can fail as the thermal environment changes [11]. However, holobiont responses are not uniform across the landscape [12] and future responses are difficult to predict, especially given that the relative contributions of the host or symbiont (or their interaction) to the bleaching phenotype are unclear.

Although species in the family Symbiodiniaceae are difficult to delineate, recent studies suggest that symbiont populations can respond to mosaics of environmental conditions across a wider range of spatial scales than their hosts. The Symbiodiniaceae family has recently been described and now formally recognizes as genera what were previously 'clades' that were defined by genetic and functional differences [13]. Thus, previous studies comparing differences between clades are now recognized as comparisons between genera. However, differences between Symbiodiniaceae *species* (previously within-clade) can result in the specialization on varying temperature and light conditions [14], as well as host species [15]. Almost all of the evidence for environmental specialization in this group comes from comparisons *between* genera [16], which show performance differences across environments that vary on scales ranging from centimetres [14] to hundreds of kilometres [17]. The few studies examining within-species diversity reveal patterns consistent with low levels of gene flow between populations separated by hundreds of kilometres [18]. No study has identified genetic differentiation or adaptive advantages within populations across shorter distances with multilocus datasets, and thus the minimum distances over which symbiont populations can locally adapt are unknown.

Testing for local adaptation or coevolution across variable environments with high species diversity such as coral reefs is difficult due to the large number of potentially interacting host and symbiont populations. However, three congeneric species of sea anemone, *Anthopleura elegantissima*, *A. xanthogrammica* and *A. sola* inhabit the highly heterogeneous intertidal zone along the west coast of North America and are the only hosts of the microalgal endosymbiont, *Breviolum 'muscatinei'* [10,19]. Multi-scale selection mosaics arise due to spatial and temporal variation within the intertidal zone (one to tens of metres) and across the geographical ranges (thousands of kilometres) of *Anthopleura* spp. Most notably, temperature and light gradients can greatly vary across latitude and depth in the intertidal zone. All three anemone species are sympatric from Bodega Bay, CA to Pt. Conception, CA and in upwelling zones south of Pt. Conception. *Anthopleura sola* and *A. xanthogrammica* are both large, solitary species that live primarily lower in the intertidal zone, extending down to 15 m into the sub-tidal zone [20]. All three host species are gonochoric broadcast spawners, with planktonic larval durations on the order of several weeks [21]. Consistent with their larval dispersal potential, previous population genetic studies using a limited number of loci concluded that populations of *Anthopleura* along the Pacific coast are panmictic [22]. However, more recent studies have shown that *A. elegantissima* exhibits population structure across its geographical range but high rates of gene flow due to long planktonic larval stages prevent locally beneficial alleles from increasing in frequency across the geographical range at most, but not all, host genomic loci [23]. Previous studies using a limited number of loci across a small geographical range revealed symbiont differentiation in *B. 'muscatinei'* across smaller spatial scales than their host populations of *A. elegantissima* [24], suggesting that host and symbiont populations face different evolutionary constraints in terms of matching local environmental conditions.

In this paper, we use a population genetic approach to identify the spatial scales over which host and symbiont populations could match selectively important environmental variation. First, we examine the degree to which differentiated symbiont populations are associated with varying environmental conditions at the macroscale of the geographical range and the mesoscale of the intertidal zone (using the surrounding benthic community as a proxy for environmental variation at this scale). Second, we test the hypothesis that microscale host–symbiont interactions shape the genetic structure of the partners, consistent with the idea that host morphology or cellular interactions determine which partnerships are successful. We begin addressing this hypothesis by identifying covarying regions of the host and symbiont genome, combined with analyses designed to identify whether host species-level effects or the genotype of each individual host drives these associations.

## 2. Material and methods

### (a) Field collection

We sampled populations of *Anthopleura* sp. and *B. 'muscatinei'* in the southern portion of the geographical ranges of these species, from Cape Blanco, OR (USA) to the southern tip of Baja California, Mexico. Additionally, we included populations of *A. xanthogrammica* from more northerly locations up to Washington State to more fully characterize the population genetic structure of this species through a larger portion of its range. The distributions of the three host species largely overlap across this range (the continuous range of *A. xanthogrammica* extends south to Pt. Conception, CA but occasionally individuals occur in areas of upwelling near Ensenada, Mexico; [25]), where they span highly heterogeneous conditions due to pockets of upwelling that bring cooler, nutrient-rich deep water to the surface [26]. We sampled nine sites, six in Baja California, Mexico, two in California and one at Cape Blanco, OR. At each site, we sampled 15–20 individuals of each host species, when present (electronic supplementary material, table S1). We photographed and recorded the GPS coordinates of each sampled individual, noted whether the polyps were submerged or exposed to air during low tide, and qualitatively characterized the surrounding benthic community (seaweeds, algae and sessile marine invertebrates). We preserved two sets of tentacle clippings from each sampled host anemone in 20% DMSO/0.5M EDTA solution and 80% ethanol for downstream genetic analysis.

### (b) DNA extraction and sequencing

We extracted DNA from the preserved host tentacles using a standard CTAB extraction procedure (see electronic supplementary material for a detailed description of extraction and library preparation protocols). To increase the representation of symbiont reads in the final library, we enriched samples for symbiont cells using a Cytomation MoFlo high-speed cell sorter (Beckman). Gates for each sort were set to keep cells whose fluorescence corresponded to excitation of chlorophyll generated from 488 nm to 635 nm wavelengths. We prepared uniquely barcoded RAD-seq libraries (see electronic supplementary material, table S1 for final sample sizes for all host species and symbiont populations by location) for all samples and sequenced them on the Illumina 4000 platform.

### (c) Analysis

We evaluated population genetic patterns in the hosts across their geographical ranges, and symbionts across a set of nested spatial scales (see electronic supplementary material for details). Across the geographical range, we characterized isolation by distance (IBD) in the hosts using pairwise $F_{ST}$ values. For the two

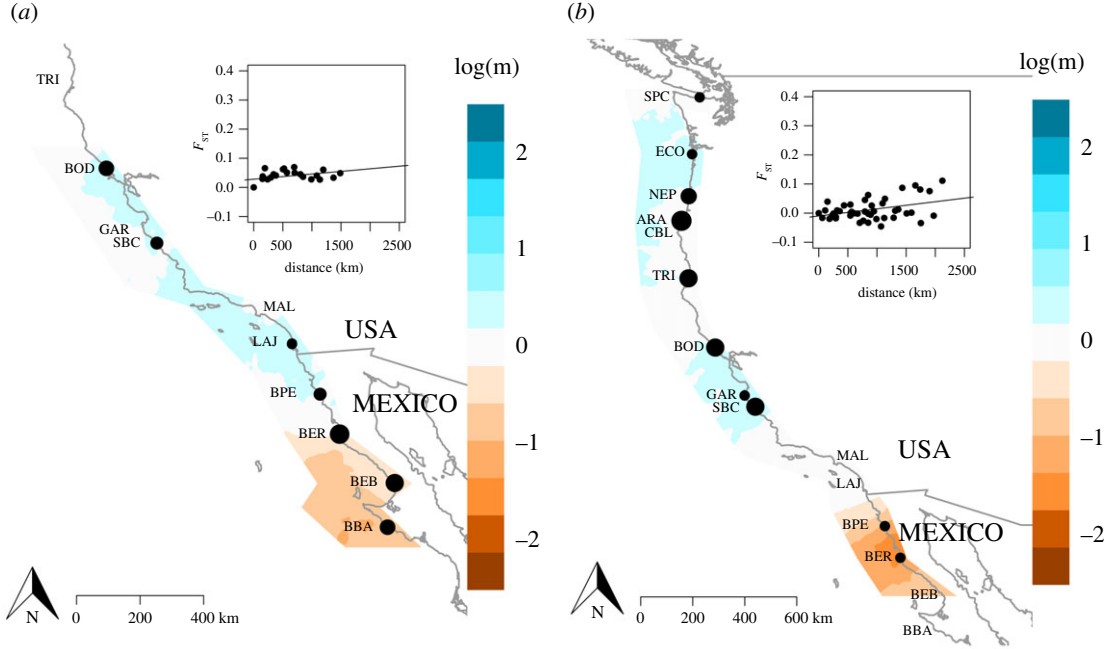

**Figure 1.** *eems* analysis of relative rates of gene flow in *A. sola* and *A. xanthogrammica*. Colours represent geographical regions where gene flow is more (blue) or less (red) likely on a log scale, with colour intensity denoting the magnitude of gene flow (darker blue shading corresponds to higher values of gene flow, darker red shading corresponds to lower values of gene flow). Points at each sampling location are scaled to sample size (note, point is absent but location label is present for locations from which no samples were collected). Insets show the range-wide isolation by distance relationship of genetic differentiation ($F_{ST}$) by geographical distance. (*a*) *A. sola*, and (*b*) *A. xanthogrammica*. (Online version in colour.)

host species, we further evaluated geographical deviations in gene flow from IBD patterns using the software *eems* [27]. For the symbiont populations, we characterized the genetic structure using principle components analysis. We assessed genetic differentiation across smaller spatial scales by developing a proxy for the microhabitat of each polyp using the surrounding biotic community (see electronic supplementary material for details) and used this as an environmental measure in a principle components analysis, a partial Mantel test, as well as using the R package BEDASSLE [28] to identify the relative contributions of geographical and environmental distance in structuring populations. Finally, we estimated the covariance between all host and symbiont loci at the individual level (i.e. blind to geographical location or host species), and ran a linear model on the allele frequencies at each symbiont locus with host species as a fixed effect. We then re-calculated the covariance after removing the host species effect on symbiont allele frequencies to quantify the role of species-specific interactions in shaping covariance between host and symbiont loci.

## 3. Results

### (a) Host

All three host species were present and common at geographical locations within their previously described contiguous geographical ranges. As expected, *A. xanthogrammica* was rare south of Pt. Conception, but was present at BPE and BER, which are geographical locations dominated by stronger upwelling regimes [26]. None of the three host species were found at sites south of BBA on the Baja California Peninsula.

After quality filtering, we retained 16 917 SNPs across 158 individuals for the three anemone species we genotyped for this study (electronic supplementary material, table S1). Range-wide patterns of genetic diversity ($\pi$) are highest for *A. sola* (0.0095), and are much lower for *A. xanthogrammica*

(0.0025); for reference, in *A. elegantissima* populations $\pi = 0.0035$ [23]. There is virtually no detectable signal of IBD in *A. sola* with low levels of genetic differentiation ($F_{ST} = 0.021$–0.049) between populations. Conversely, *A. xanthogrammica* populations show IBD, largely due to high levels of differentiation between the three individuals we sampled in Baja California, Mexico and the contiguous populations north of Pt. Conception, CA (see electronic supplementary material, tables S2–S4 for all pairwise $F_{ST}$ values). The overall slope of the best-fit regression of scaled $F_{ST}$ to geographical distances is $1.79 \times 10^{-5}$ for *A. sola* ($p = 0.0047$) and $2.45 \times 10^{-5}$ for *A. xanthogrammica* ($p = 1.71 \times 10^{-5}$; figure 1). ADMIXTURE found the most support for $K = 1$ for both *A. sola* and *A. xanthogrammica*.

*A. sola* and *A. xanthogrammica* both exhibit reductions in gene flow in the southern portions of their geographical ranges relative to the northern locations. Strikingly, the *eems* analyses, which account for deviations in IBD relative to the range-wide signal, suggest that *A. sola* and *A. xanthogrammica* show limited levels of gene flow in the southernmost portions of their ranges (figure 1). In both *A. xanthogrammica* and *A. sola*, BAYESCAN failed to identify any loci putatively under positive selection.

### (b) Symbiont

The 1470 SNPs that were successfully genotyped and passed quality filters show genetic structure within and between host species and geographical sites (electronic supplementary material, table S1). After excluding hosts whose symbionts exhibited low genotyping rates, 56 individuals remained in the final dataset. Our initial dataset contained 2192 SNPs, which was reduced to 1470 after pruning linked loci. Populations at each geographical location we sampled (regardless of the host species in which they were found) exhibit high levels

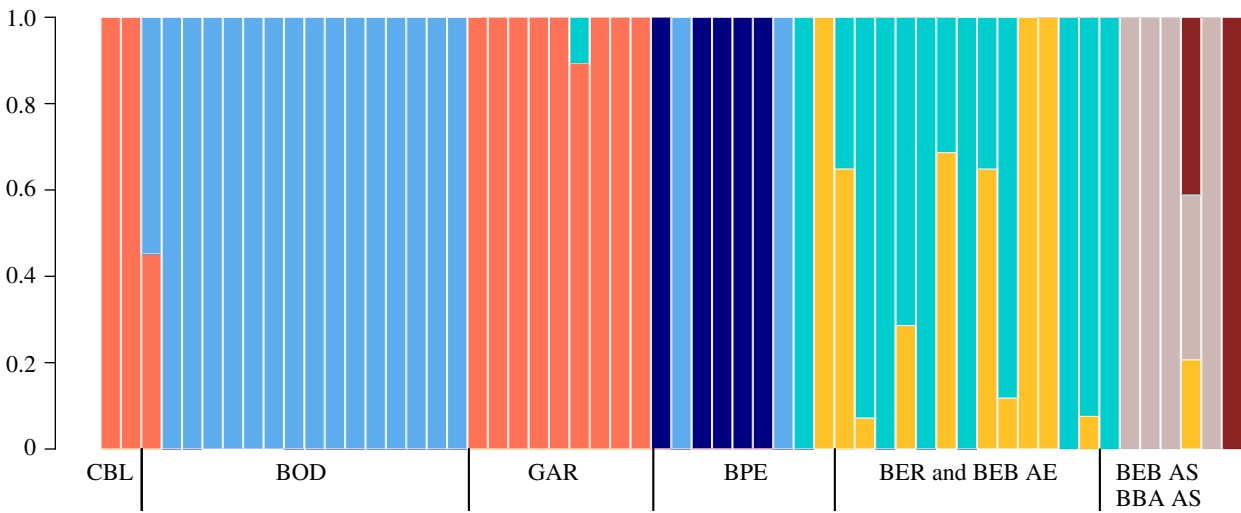

**Figure 2.** ADMIXTURE analysis of *B. 'muscatinei'* populations. The proportion of the genome assigned to each subpopulation (represented by different colours) in each individual is denoted on the *Y*-axis. From left to right, individuals are arranged from north to south. Unless otherwise noted, site labels encompass symbionts from all host species present at that site. As estimated by the lowest coefficient of variation, the most likely number of subpopulations (*K*) = 7. (Online version in colour.)

of $F_{ST}$ (electronic supplementary material, table S5), and ADMIXTURE best supports a model of seven subpopulations of symbionts across the geographical range we sampled (*ca* 2000 km). Four of these subpopulations correspond to geographical location: Bodega Bay, CA (BOD), Big Sur, CA (GAR), Ensenada, Baja California (BPE) and the southern-most populations (BER, BEB and BBA) of Baja California (figure 2). However, the final subpopulation, composed of individuals at the two southernmost sites (BEB and BBA), distinguishes the symbiont populations of *A. sola*. Interest-ingly, the symbiont population of *A. elegantissima* at these southern-most sites fails to map to the *B. minutum* genome (or any other published genome available on the NCBI data-base or reefgenomics.org). These samples are omitted from further discussion due to a lack of homologous loci between them and all other samples.

The PCA also distinguishes symbiont populations inhabiting the southern-most sites, where the first principle component separates southern *A. sola* symbiont populations (BBA and BEB) from all other symbiont populations (figure 3). The axis of variation on PC2 separates symbiont populations from BOD and BPE from every other geographical location (figure 3). Although these two locations are separated by thousands of kilometres, they are both regions of intense upwelling. A neighbour-joining tree with bootstrapped resam-pling to quantify node support also revealed the geographical structure and population similarities similar to those revealed by the ADMIXTURE and principal components analyses (electronic supplementary material, figure S1).

The AMOVA shows a hierarchical structure in the genetic differentiation of the symbionts across their geographical range and within host species. Overall, geographical location explains the largest percentage of genetic variation between symbionts ($\varphi = 0.1468$, $p = 0.002$), followed by host species ($\varphi = 0.0612$, $p = 0.041$).

The benthic community composition (and, by inference, environmental variation) across the intertidal zone correlates with changes in symbiont genotypes within and between sites. Environmental patterns vary within and between sites: PC1 of the benthic community surrounding each

individual is correlated with PC2 of the symbiont genetic dataset ($r^2 = 0.2602$, $p = 8.127 \times 10^{-5}$) (figure 3). Partial Mantel tests on the correlation between symbiont genetic dis-tance and benthic community ecological distances reveal a weak correlation, when controlling for geographical distances separating sites ($r = 0.1183$, $p = 0.03$). The results from BEDAS-SLE revealed an aE : aD (relative impact of environmental distance to geographical distance on genetic covariance between individuals) ratio of 0.0015. To contextualize this ratio, mean environmental distances between individuals from the same geographical sampling site with relatively large environmental distances (BOD) and relatively small environmental distances (BER) were 11.54 and 6.78, respect-ively. These mean benthic community distances result in the same correlations in allele frequencies between the sym-bionts found within two individuals as approximately 4500–7500 km of geographical separation.

Finally, we attempted to identify covarying loci across host and symbiont genomes. Overall, based on an analysis that includes all individuals regardless of host species, the LD between the hosts and their associated symbionts is low and, in most cases, near 0. The patterns of LD decrease towards no covariation when the effect of host species is regressed out of the symbiont allele frequencies (figure 4), from a mean absolute value of 0.034 to a mean of 0.019.

## 4. Discussion

Geographical mosaics can impose highly variable selection pressures on host–microbial partnerships; the rate and magnitude of response, in turn, is shaped by transmission mode, gene flow and generation times in both partners [29,30]. In the *Anthopleura* spp.–*B. 'muscatinei'* partnership, a combination of low rates of gene flow and strong selective gradients across metres in the intertidal zone as well as across their geographical range shape the genetic structure of the symbiont. The opposite is true in the hosts, where populations of all three species of *Anthopleura* exhibit high levels of gene flow across their geographical ranges, reducing

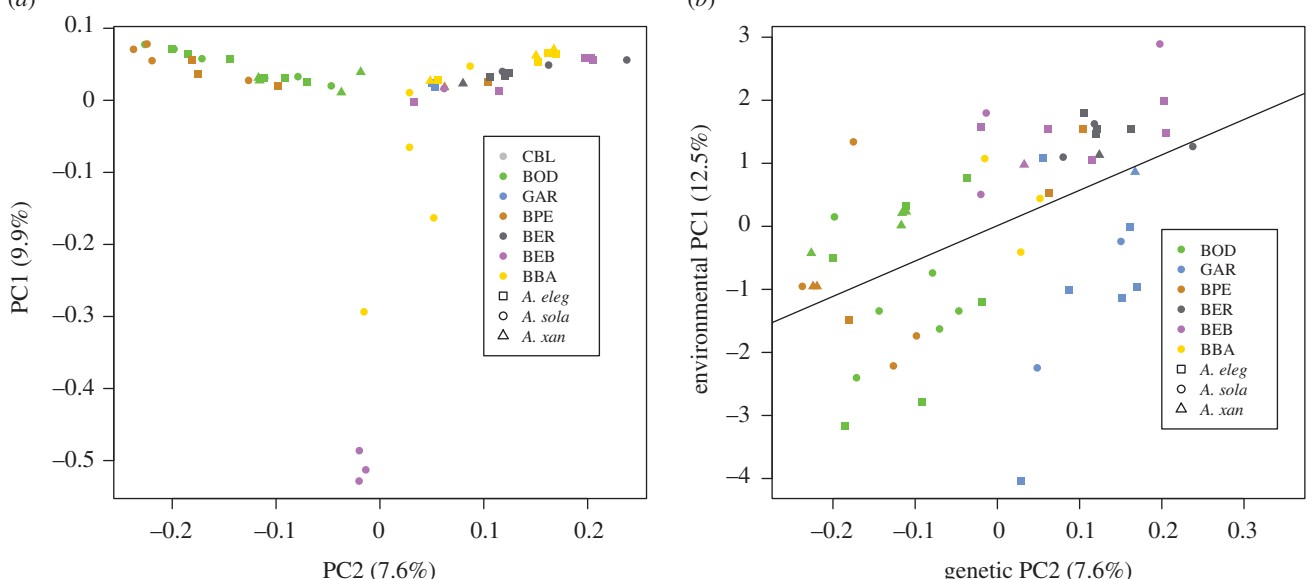

**Figure 3.** (a) Major axes of genetic variation in symbiont populations. PCA results from the symbiont populations along the Pacific coast of North America. Each site is coded by colour, each shape represents symbiont populations from the three host species. (b) Correlation of genetic and environmental variation in symbiont populations. This plot shows the correlation between PC1 of the environmental benthic community dataset regressed against PC2 of the symbiont genetic dataset ($p = 8.13 \times 10^{-05}$, $R^2 = 0.2602$). (Online version in colour.)

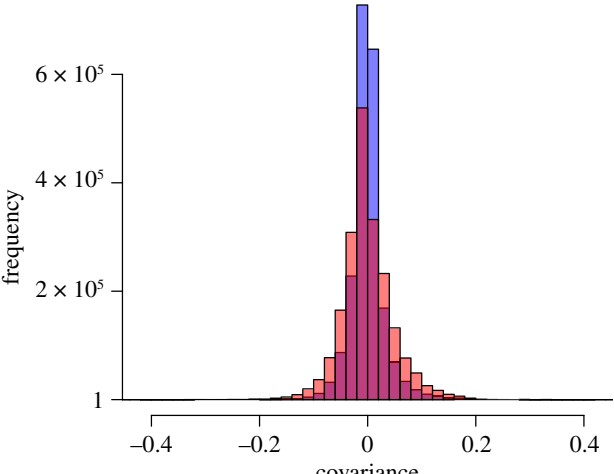

**Figure 4.** Covariance between host and symbiont loci. In red, a histogram of linkage disequilibrium values calculated for all host loci against all symbiont loci across all populations included in this study. In blue, the same set of loci where the host species contribution (as estimated using a linear model for each locus) has been subtracted out of the allele frequencies of the symbionts. (Online version in colour.)

population genetic signals of local adaptation across any of the spatial scales we examined in this study. These patterns highlight significant disparities between the spatial scales over which populations of *Anthopleura* sp. and *B. 'muscatinei'* might be able to adapt to local conditions, and suggest that symbiont populations can match variation at smaller spatial scales than their hosts. Patterns of genetic differentiation for *A. sola* and *A. xanthogrammica*, like those previously reported in *A. elegantissima* [23], imply that gene flow is high across much of their geographical ranges, only becoming limited towards the southern range boundaries of all three species. The symbionts, on the other hand, exhibit a large degree of genetic variation, partitioned across macro- (the geographical

range), meso- (the intertidal zone) and microscales (the internal host environment).

## (a) Hosts

Similar patterns of genetic differentiation in populations of *A. xanthogrammica* and *A. sola*, combined with previous results in *A. elegantissima* [23], suggest that similar oceanographic or biological processes regulate gene flow across each species' geographical range. As in *A. elegantissima*, both *A. sola* and *A. xanthogrammica* exhibit limited gene flow in southern populations relative to expectations based on IBD (figure 1), although the absolute levels of IBD in the two solitary congeners are lower than in *A. elegantissima* [23]. The large stretch of coastline that is unfavourable habitat for *A. xanthogrammica* between Pt. Conception and the northern region of Baja California could explain this pattern, as we found very few individuals of this species in this region. However, the distribution of *A. sola* is continuous throughout the same range and we observed that they were highly abundant at all locations south of Pt. Conception, which suggests that the population genetic patterns we found in this study are shaped by a universal set of factors acting on all three host species.

One explanation for the increased IBD signal in *A. elegantissima* is that dispersal of *A. elegantissima* is not as common or extensive as in *A. sola* or *A. xanthogrammica*, possibly due to decreased larval duration, survival in the plankton, or lower population densities, although no study has definitively shown this to be the case. Alternatively, *A. elegantissima* could experience more post-settlement selection than its two solitary congeners, although scans of $F_{ST}$ outliers find little evidence for selection across its geographical range [23]. Finally, only a small amount of the host genetic variation (PC4, proportion of genetic variance explained = 2.7%) is correlated with the composition of the local benthic community surrounding each polyp (electronic

supplementary material, figure S4). PC4 distinguishes individuals from BER and BEB (these locations are past the southern range limit of *A. xanthogrammica*). However, analysis of the populations in ADMIXTURE and *eems* fails to reveal any evidence for subpopulation structure based on biotic (and, presumably, physical) aspects of microhabitat, as in coral species whose ranges span depths with dramatically different illuminance [31,32]. These results suggest that local adaptation is more likely in southern populations due to lower rates of gene flow, which could allow those populations to specialize on the conditions specific to the southern portions of their geographical ranges.

## (b) Symbionts
### (i) Macroscale—the geographical range

Previous studies demonstrated that at the inter-genus (occasionally intra-genus) level, symbiont lineages may exhibit genetic structure at the level of ocean basins [33], biogeographic provinces [34], and host species [4,15,35,36]. At the largest spatial scales, and consistent with studies using microsatellite markers [15,37], symbiont populations in *Anthopleura* spp. along the Pacific coast are highly differentiated by geographical location, with some of the greatest differences occurring between neighbouring locations (e.g. BPE and BER). Previous studies of symbionts inhabiting *A. elegantissima* along the Californian coast detected similar patterns using three genetic loci, with major genetic breaks at Cape Mendocino and Monterey Bay [24]. Similar to the interpretations in [24], the evidence for population structure between geographical locations in this study implies that gene flow between symbiont populations is likely much lower than in the hosts, or that within-generation selection is much stronger in the symbionts. These patterns are similar to within-genus symbiont patterns found in corals across the Caribbean [18], which show genetic structure by geographical location and light environment. Notably, Pt. Conception, which coincides with a transition between biogeographic provinces [38], does not divide subpopulations of *Breviolum 'muscatinei'*, as is the case in the host *A. elegantissima* [23]. Signals of population substructure at scales of several hundreds of kilometres are at a shorter spatial scale than the transition in biogeographic provinces, and require other explanations.

Symbiont populations are more genetically similar when they originate from locations sharing comparable benthic communities, a proxy for environmental similarity, even when those locations are thousands of kilometres apart. For example, the Bodega population in northern California and the Ensenada population in Baja California, Mexico share genetically similar symbiont populations and a similar benthic community (notably these populations occur on either side of the transition between the Californian and Oregonian biogeographic provinces). Cold, upwelled waters are a dominant abiotic factor in both of these locations and act on the scale of hundreds of kilometres [26,39], similar patterns have also been described in tropical and subtropical Symbiodiniaceae species that exhibit increased genetic differentiation between tropical and subtropical populations, despite vertical transmission in host larvae where long-distance dispersal between the two habitats is likely to be common [40]. The fact that correlations between principle components explaining genetic and environmental variation are evident when comparing populations in geographical locations separated by thousands of kilometres suggests that *B. 'muscatinei'* populations draw on the same genetic variation to match environmental variation across the geographical range of *Anthopleura* spp.

### (ii) Mesoscale—the intertidal zone

The intertidal zone is a highly variable environment dominated by environmental stress, competition and predation that collectively structure dramatically different biotic communities over the scale of metres [41,42]. The shifts in abiotic conditions, notably light, desiccation and thermal gradients, that result in large changes to the biotic community could create a fine-scale geographical mosaic that shapes symbiont populations, but has little to no effect on host populations. This could be due to shorter generation times in the symbiont, allowing selection to act more rapidly than in their hosts. Alternatively, gene flow and migration could be lower in the symbionts, potentially leading to differentiation across depth in the intertidal zone. Clear shifts in symbiont species arise due to depth in coral reef environments [43], and symbiont populations at the leading edge of geographical range expansions exhibit reduced genetic diversity consistent with low rates of gene flow and local adaptation [44], although no study to our knowledge has shown similar *intra-specific* patterns and variation at the level of populations on which selection might act.

This study capitalizes on biotic shifts across geographical sites as well as depth in the intertidal zone to create an environmental proxy, which, in turn, is correlated with major axes of symbiont genetic variation. Environmental variation could favour symbiont strains that specialize on a certain location in the intertidal zone. For instance, variation in light regime leads to shifts in the dominant symbiont genus across scales as small as a single coral head [14]. Previous studies of symbiont population structure focusing on geographical locations separated by tens of kilometres to thousands of kilometres (e.g. [24,32,45,46]) may not have detected fine-scale structure in symbiont populations because they lacked the genetic power to resolve ecologically significant genetic variation across the genome. Furthermore, although marine environments are highly heterogeneous, sampling from a single geographical location without taking local heterogeneity into account will obscure any genetic signal arising due to environmental variation across metres of depth, leading to high within-site variance that appears uncorrelated with environmental shifts over thousands of kilometres. This is especially true in the intertidal zone, where variation across vertical scales of less than a few metres can lead to aerial exposure for hours during a low tide for individuals in the upper intertidal zone compared to individuals inhabiting permanent pools that are completely submerged for their entire life.

Further evidence from this study for how environmental variation shapes symbiont genetic/population structure comes from BEDASSLE, which controls for spatial autocorrelation in environmental variables. This analysis shows that the average variation within a local environment (across approx. 5 m of depth) results in allele frequency correlations of the same magnitude as those that arise between populations separated by thousands of kilometres. This appears to occur independently at most geographical locations, as the results from the ADMIXTURE analysis suggest that

overall rates of gene flow across the geographical ranges of these species are relatively low, a pattern apparent in other studies of symbiont populations separated by hundreds of kilometres to thousands of kilometres of km using neutral microsatellite markers [15]. Local beneficial alleles could rapidly increase in frequency in the symbionts, where rates of division for other genera in the family Symbiodiniaceae in culture is on the order of days to weeks [47], allowing selectively advantageous strains to proliferate quickly at a geographical location or in a particular microhabitat

The overall pattern of genetic structure across the geographical range we sampled in this study, combined with fine-scale matching of symbiont genetic diversity to a local biotic community suggests that the genetic diversity needed to match environmental variation across depth in the intertidal zone is present in the symbiont population at any one geographical location. Alternatively, symbionts may extensively disperse and come to dominate a single geographical location, although dispersal is unlikely to happen while associated with either sessile host adults or dispersing larvae, as symbionts are horizontally transmitted in *Anthopleura* [48]. This explanation requires sufficient amounts of neutral divergence (e.g. due to genetic drift during a founder event) to cause detectable genetic structure across the geographical range we found in the ADMIXTURE analysis. However, founder events resulting in drift would also eliminate genetic variation, which we would expect to reduce the ability of symbionts to match local environmental variation (tens of metres), the kind of association revealed by the BEDASSLE analysis. The structure resulting from highly variable environments could also contribute to high genetic variance in symbiont populations, variation in partner quality across environments and selection for optimal partnerships [11]. This study has shed light on the relevant spatial scales over which these hypotheses should be addressed. For example, clonal members of *A. elegantissima* can be reciprocally transplanted across the intertidal zone to assess the rate at which genetically identical hosts acquire or alter symbiont strains that match their environment.

### (iii) Microscale—host–symbiont interactions

Genetic differences that allow symbiont populations to match local conditions can arise from the specific microenvironment a host creates (either internally or due to the niche that a host species occupies in the intertidal zone) or from interactions between loci in the host and symbiont genomes. We found that symbiont populations are genetically structured by host species within sites and that inter-species covariation (linkage disequilibrium) is partly explained by allele frequency differences in symbiont populations inhabiting the three different host species (figure 4). These patterns could reflect locally adaptive differences between symbiont strains that correspond to variation in thermal, light or desiccation stress associated with the height at which the host lives in the intertidal zone [41], the internal host microenvironment (e.g. variation in ectodermal thickness influencing light transmission to symbionts; [10]), or cellular interactions between hosts and symbionts that establish and maintain the endosymbiotic partnership [49]. At the southern-most sites (BBA and BEB), symbiont populations in *A. sola* are so distinct that they explain most of the variation apparent

from the PC analysis, outweighing other patterns associated with abiotic conditions or geographical distance between populations revealed in the second principle component. Furthermore, the *A. elegantissima* population at BBA hosted symbionts, but these failed to align to the *B. minutum* genome (or any other published symbiont genome), suggesting that they are the most genetically distinct population we sampled and likely represent another species, if not genus.

Host-specificity in the southern portion of the geographical range could result from historical or contemporary forces. One explanation for symbiont segregation by host species is historical isolation leading to coevolution, where a single isolated host species at this location tightly coevolved with the local symbiont population, leaving the symbiont partner unable to interact with other host species when the host populations subsequently became sympatric [50,51]. However, our analyses failed to uncover any evidence for co-isolation in the host and symbiont populations that would lead to coevolution. To the contrary, although there is some evidence of reduced gene flow in the southern portion of the geographical ranges of the host species, there is no evidence for population substructure reflecting historical isolation. Alternatively, the abiotically challenging habitats at the southern-most range limits of *A. sola* and *A. elegantissima* could favour symbiont strains that specialize on the morphological attributes of host polyps (e.g. ectodermal thickness, size of polyp) or the host species-specific variation in microhabitat occupation (e.g. location in the intertidal zone, local levels of irradiance, temperature and wave exposure). Similar beneficial pairings arise in non-nutritional partnerships where symbionts and hosts facilitate each other's persistence in harsh and stressful conditions [52], even if those conditions reduce the benefit each partner provides to the other in the interaction [53]. A key next step is to determine if partner specificity arises due to stressful conditions, which could become more common as marine populations face rapidly increasing thermal stress due to global warming.

The genetic structure of *Anthopleura* sp. and their associated Symbiodiniaceae populations across large and small spatial scales both affects and reflects how natural selection and coevolution shape the performance of the holobiont. Across species within the Symbiodiniaceae family, patterns consistent with local adaptation [17] and with competitive exclusion between symbiont strains [14] suggest that local environmental conditions contribute to the diversity of species and the distribution of genetic variation within the Symbiodiniaceae. These studies further show that genetic differences *between* Symbiodiniaceae genera are related to their relative performance across different thermal and irradiance regimes. Within the vertical span of the intertidal zone, *Anthopleura* and their symbionts can experience greater variation in temperature and light than what most sub-tidal tropical species experience across their entire geographical ranges. The populations of *Breviolum 'muscatinei'* in this study have genetic patterns consistent with the hypothesis that genetic variation *within* symbiont species (as defined by *ITS2* and organellar loci) can allow symbionts to match their local abiotic environment, as well as the intra-cellular host biotic environment. However, analysing *ITS2* alone would fail to uncover some of these patterns, notably those across the intertidal zone at a single geographical location

as well as covariation between host and symbiont loci. Population genetic patterns in the hosts suggest that gene flow likely swamps any multi-generational selective response to environmental heterogeneity across the intertidal zone. More importantly, adaptive benefits could falsely appear to arise from host populations if studies fail to take the genetic variation of the symbionts into account. Future work should experimentally test whether these genetic patterns actually reflect the enhanced performance of the symbiont (and possibly the holobiont) in its local environmental conditions, given that some symbiont strains could be parasitic in some environmental contexts [52], and that the partnership responds to fluctuations in the host nutritional budget differently, depending on the host species [54]. Fully characterizing the costs and benefits of the partnership across multiple scales is a crucial next step which can be used to test hypotheses that explain how or if the relationship remains stable over time, to reveal the genetic basis of locally adapted traits, and to predict how host and symbiont populations will respond (independently and as partners) as the climate changes.

Ethics. This research was carried out under permits from the State of California (12606), Oregon (18070), Washington (14-242) and with permission to collect in Mexico for L.H.

Data accessibility. Sequencing data are available on the NCBI Sequence Read Archive (BioProject PRJNA698612), final genotype datasets are available from the Dryad Digital Repository: https://doi.org/10.25338/B8WK8F [55].

Authors' contributions. L.H. carried out fieldwork, reviewed and edited the manuscript. B.H.C. carried out field and laboratory work, performed analyses, wrote the original draft as well as reviewed and edited of the final manuscript.

Competing interests. We declare we have no competing interests.

Funding. This work was supported by grants from UCMEXUS to Richard Grosberg and L.H., NSF (OCE-1459815) and the University of California Agricultural Experiment Station to Richard Grosberg, the UC Davis Natural Reserve System and the Center for Population Biology at UC Davis. Cell sorting was undertaken at the University of California Davis Flow Cytometry Shared Resource Laboratory with funding from the NCI (grant no. P30CA093373) and the NIH (grant no. S10 OD018223).

Acknowledgements. The authors would like to acknowledge valuable feedback on this manuscript from Richard Grosberg, Graham Coop and Rob Toonen. The authors would like to also acknowledge Bridget McLaughlin and Jonathan Van Dyke for their technical assistance in sorting symbiont cells.

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
