## [Peer Review File · Proceedings of the Royal Society B: Biological Sciences]

Review History

RSPB-2020-2896.R0 (Original submission)

Review form: Reviewer 1

Recommendation

Accept with minor revision (please list in comments)

Scientific importance: Is the manuscript an original and important contribution to its field?

Excellent

General interest: Is the paper of sufficient general interest?

Good

Quality of the paper: Is the overall quality of the paper suitable?

Excellent

Is the length of the paper justified?

Yes

Should the paper be seen by a specialist statistical reviewer?

No

Do you have any concerns about statistical analyses in this paper? If so, please specify them explicitly in your report.

No

It is a condition of publication that authors make their supporting data, code and materials available - either as supplementary material or hosted in an external repository. Please rate, if applicable, the supporting data on the following criteria.

Is it accessible?

No

Is it clear?

N/A

Is it adequate?

N/A

Do you have any ethical concerns with this paper?

No

Comments to the Author

Summary

Cornwell & Hernandez analyzed the population genetics of congeneric sea anemones and their shared endosymbiont along the west coast of North America using a RADseq approach. They find little population structure in the hosts, instead identifying an isolation-by-distance pattern. But the symbionts exhibit more structure, with at least seven populations across the biogeographic range. Genetic structure in the symbiont is influenced by macroscale (geography), mesoscale (intertidal zonation), and microscale (host specificity) processes. Given the symbiont's shorter generation time and greater structure (due to low gene flow and/or strong selection), the authors conclude that the symbionts are better positioned to adapt to climate change, which may influence the adaptive capacity of the holobiont.

This is an excellent manuscript: thoughtful, easy to read, and interesting. The methods are sound and the conclusions follow from the data. I wouldn't say the results are particularly surprising (many of the patterns have been identified previously, albeit among symbiont species, not within species), but there are few studies in this intra-species space and even fewer that take advantage of high-throughput approaches to look at population structure. This focus on within-species variation does provide novel insight into the relative importance of coevolution among partners. I was particularly impressed with the analyses of correlations between genetic and environmental variation in the symbiont, as well as covariance between host and symbiont loci. I have not seen such efforts in the Symbiodiniaceae literature before; I hope this manuscript will popularize them. The Anthopleura-Breviolum association is a fantastic system; it's fairly unique in that it allows for consideration of macro/meso/micro-scale processes simultaneously. I really don't have much to critique here—just a few suggestions for minor improvements. Great job.

Minor Comments

You should probably include the Symbiodiniaceae revision in the introduction, as it provides context for the within/between genus discussion. LaJeunesse et al. 2018 Current Biology ("Systematic revision of Symbiodiniaceae highlights the antiquity and diversity of coral endosymbionts").

I found two instances the taxonomic rank should be adjusted. At L301: change "within-clade" to "within-genus." At L447: change "within symbiont genera" to "within symbiont species"

(because you are talking about variation within *B. muscatinei*; because the statement holds for intra-specific variation; and because ITS2 is a species-level marker, though it also gives you the genus at the same time).

I found the Supplemental Methods to be quite helpful for understanding the analyses and they're only two pages long – if space is not an issue, I would recommend adding them back in. I realize Proceedings has a pretty stringent page limit. The discussion is a bit long, so it might be worth trying to cut it down somewhat to make more space for the methods. You might also condense figures to free up space. Figs 1 and 2 (and possibly 3) could be combined, as could Figs 4, 5, and 6.

When discussing the BBA *A. elegantissima* symbiont population whose reads did not align with the *B. minutum* genome, it is worth mentioning that this likely represents a unique species, rather than just a distinct population of *B. muscatinei*.

For the paragraph on specific environments favoring particular symbiont strains (L413-423), consider incorporating Grupstra et al. 2017 Coral Reefs ("Evidence for coral range expansion accompanied by reduced diversity of Symbiodinium genotypes"), as it's relevant.

Review form: Reviewer 2

Recommendation

Accept with minor revision (please list in comments)

Scientific importance: Is the manuscript an original and important contribution to its field?

Good

General interest: Is the paper of sufficient general interest?

Good

Quality of the paper: Is the overall quality of the paper suitable?

Good

Is the length of the paper justified?

Yes

Should the paper be seen by a specialist statistical reviewer?

No

Do you have any concerns about statistical analyses in this paper? If so, please specify them explicitly in your report.

No

It is a condition of publication that authors make their supporting data, code and materials available - either as supplementary material or hosted in an external repository. Please rate, if applicable, the supporting data on the following criteria.

Is it accessible?

Yes

Is it clear?

Yes

Is it adequate?

Yes

Do you have any ethical concerns with this paper?

No

Comments to the Author

This manuscript is an original study tracking genetic changes in symbiotic dinoflagellates over a latitudinal gradient. Its findings and the Authors' conclusions are consistent with their data and related studies as well as biological theory. Moreover, this work sets the bar higher for future approaches in how investigators ought to study symbiotic animals and their mutualistic microbes. Their most significant finding, that the symbiont is more evolutionarily responsive to selection from external environmental pressures, is substantiated by the phylogenetic differences where most symbiont lineages have evolved recently vs. the older longevity apparent in many host lineages. This is a solid contribution and with some revision should make for a nice publication.

The Sanders and Palumbi 2011 paper provided a nice study showing many of the genetic differences in the *Breviolum* symbiont found in *Anthopleura* along the coastline from Washington to California. I would like to see more effort to reconcile the present work with this paper. From the preliminary genetic work from this earlier paper, it appears that the northern population in Washington and Oregon may be a different species than that found southern California (and possibly a third). I think that the present work would have benefited from providing at least one phylogenetic marker (LSU?) to help provide a phylogenetic framework for a better perspective of their findings. This would also reveal the identity of the symbiont found in the most southern anemones. Clearly that symbiont is of another genus.

There are a few papers that also show how the genetics of the host does not match with that of the genetic structure of the symbiont, please read and include this in your citations:

Baums, I.B., Devlin-Durante, M.K., and LaJeunesse, T.C. (2014). New insights into the dynamics between reef corals and their associated dinoflagellate endosymbionts from population genetic studies. *Mol. Ecol.* 23, 4203-4215.

Pettay, D.T., and LaJeunesse, T.C. (2013). Long-range dispersal and high-latitude environments influence the population structure of a "stress-tolerant" dinoflagellate endosymbiont. *PLoS One* 8, e79208.

I think that the discussion could be shortened as it is redundant in a few places and too speculative in other sections. When possible, please limit speculation to subjects where there is sufficient supporting data to warrant consideration.

Breviolum muscatinei is presently not a valid species as there is no formal characterization. So it should be written in this paper as: *Breviolum* 'muscatinei'

A thorough systematic treatment is needed for this dinoflagellate to determine if it indeed constitutes one species over the entire geographic range of the U.S. coastline or comprises two or more.

Please delete the Stat et al. 2008 paper in PNAS, cited at the end of the discussion. This is a misleading work and erroneous at all levels from the experimental design, data collection, to interpretation of results. It is a poster child of how to do bad science and get it published in PNAS if you know a member who is willing to see it through without much review (that practice has since been discontinued for reasons such as this paper). Sorry to be so blunt.

Regards
Todd C. LaJeunesse

Decision letter (RSPB-2020-2896.R0)

19-Jan-2021

Dear Dr Cornwell

I am pleased to inform you that your Review manuscript RSPB-2020-2896 entitled "Genetic structure in the endosymbiont *Breviolum muscatinei* is correlated with geographic location, environment, and host species" has been accepted for publication in Proceedings B.

The referee(s) do not recommend any further changes. Therefore, please proof-read your manuscript carefully and upload your final files for publication. Because the schedule for publication is very tight, it is a condition of publication that you submit the revised version of your manuscript within 7 days. If you do not think you will be able to meet this date please let me know immediately.

To upload your manuscript, log into <http://mc.manuscriptcentral.com/prsb> and enter your Author Centre, where you will find your manuscript title listed under "Manuscripts with Decisions." Under "Actions," click on "Create a Revision." Your manuscript number has been appended to denote a revision.

You will be unable to make your revisions on the originally submitted version of the manuscript. Instead, upload a new version through your Author Centre.

- 1) A text file of the manuscript (doc, txt, rtf or tex), including the references, tables (including captions) and figure captions. Please remove any tracked changes from the text before submission. PDF files are not an accepted format for the "Main Document".
- 2) A separate electronic file of each figure (tiff, EPS or print-quality PDF preferred). The format should be produced directly from original creation package, or original software format. Please note that PowerPoint files are not accepted.
- 3) Electronic supplementary material: this should be contained in a separate file from the main text and the file name should contain the author's name and journal name, e.g. `authorname_procb_ESM_figures.pdf`

All supplementary materials accompanying an accepted article will be treated as in their final form. They will be published alongside the paper on the journal website and posted on the online figshare repository. Files on figshare will be made available approximately one week before the accompanying article so that the supplementary material can be attributed a unique DOI. Please see: <https://royalsociety.org/journals/authors/author-guidelines/>

4) Data-Sharing and data citation

It is a condition of publication that data supporting your paper are made available. Data should be made available either in the electronic supplementary material or through an appropriate repository. Details of how to access data should be included in your paper. Please see <https://royalsociety.org/journals/ethics-policies/data-sharing-mining/> for more details.

If you wish to submit your data to Dryad (<http://datadryad.org/>) and have not already done so you can submit your data via this link <http://datadryad.org/submit?journalID=RSPB&manu=RSPB-2020-2896> which will take you to your unique entry in the Dryad repository.

Once again, thank you for submitting your manuscript to Proceedings B and I look forward to receiving your final version. If you have any questions at all, please do not hesitate to get in touch.

Sincerely,
Dr Daniel Costa
<mailto:proceedingsb@royalsociety.org>

Associate Editor Board Member: 1
Comments to Author:
Dear Brendan and Luis

Your manuscript has been reviewed by two experts in the field. They both like your manuscript but highlight a number of points that should be addressed before the manuscript is ultimately published. I hope you will find their reviews useful in the final edit of this ms.

Warm Regards

Line

Reviewer(s)' Comments to Author:

Referee: 1

Comments to the Author(s)
Summary

Cornwell & Hernandez analyzed the population genetics of congeneric sea anemones and their shared endosymbiont along the west coast of North America using a RADseq approach. They find little population structure in the hosts, instead identifying an isolation-by-distance pattern. But the symbionts exhibit more structure, with at least seven populations across the biogeographic range. Genetic structure in the symbiont is influenced by macroscale (geography), mesoscale (intertidal zonation), and microscale (host specificity) processes. Given the symbiont's shorter generation time and greater structure (due to low gene flow and/or strong selection), the authors conclude that the symbionts are better positioned to adapt to climate change, which may influence the adaptive capacity of the holobiont.

This is an excellent manuscript: thoughtful, easy to read, and interesting. The methods are sound and the conclusions follow from the data. I wouldn't say the results are particularly surprising (many of the patterns have been identified previously, albeit among symbiont species, not within species), but there are few studies in this intra-species space and even fewer that take advantage of high-throughput approaches to look at population structure. This focus on within-species variation does provide novel insight into the relative importance of coevolution among partners. I was particularly impressed with the analyses of correlations between genetic and environmental variation in the symbiont, as well as covariance between host and symbiont loci. I have not seen

such efforts in the Symbiodiniaceae literature before; I hope this manuscript will popularize them. The Anthopleura-Breviolum association is a fantastic system; it's fairly unique in that it allows for consideration of macro/meso/micro-scale processes simultaneously. I really don't have much to critique here—just a few suggestions for minor improvements. Great job.

Minor Comments

You should probably include the Symbiodiniaceae revision in the introduction, as it provides context for the within/between genus discussion. LaJeunesse et al. 2018 Current Biology (“Systematic revision of Symbiodiniaceae highlights the antiquity and diversity of coral endosymbionts”).

I found two instances the taxonomic rank should be adjusted. At L301: change “within-clade” to “within-genus.” At L447: change “within symbiont genera” to “within symbiont species” (because you are talking about variation within *B. muscatinei*; because the statement holds for intra-specific variation; and because ITS2 is a species-level marker, though it also gives you the genus at the same time).

I found the Supplemental Methods to be quite helpful for understanding the analyses and they're only two pages long— if space is not an issue, I would recommend adding them back in. I realize Proceedings has a pretty stringent page limit. The discussion is a bit long, so it might be worth trying to cut it down somewhat to make more space for the methods. You might also condense figures to free up space. Figs 1 and 2 (and possibly 3) could be combined, as could Figs 4, 5, and 6.

When discussing the BBA *A. elegantissima* symbiont population whose reads did not align with the *B. minutum* genome, it is worth mentioning that this likely represents a unique species, rather than just a distinct population of *B. muscatinei*.

For the paragraph on specific environments favoring particular symbiont strains (L413-423), consider incorporating Grupstra et al. 2017 Coral Reefs (“Evidence for coral range expansion accompanied by reduced diversity of Symbiodinium genotypes”), as it's relevant.

Referee: 2

Comments to the Author(s)

This manuscript is an original study tracking genetic changes in symbiotic dinoflagellates over a latitudinal gradient. Its findings and the Authors' conclusions are consistent with their data and related studies as well as biological theory. Moreover this work sets the bar higher for future approaches in how investigators ought to study symbiotic animals and their mutualistic microbes. Their most significant finding, that the symbiont is more evolutionarily responsive to selection from external environmental pressures, is substantiated by the phylogenetic differences where most symbiont lineages have evolved recently vs. the older longevity apparent in many host lineages. This is a solid contribution and with some revision should make for a nice publication.

The Sanders and Palumbi 2011 paper provided a nice study showing many of the genetic differences in the *Breviolum* symbiont found in *Anthopleura* along the coastline from Washington to California. I would like to see more effort to reconcile the present work with this paper. From the preliminary genetic work from this earlier paper, it appears that the northern population in Washington and Oregon may be a different species than that found southern California (and possibly a third). I think that the present work would have benefited from providing at least one phylogenetic marker (LSU?) to help provide a phylogenetic framework for better perspective of their findings. This would also reveal the identity of the symbiont found in the most southern anemones. Clearly that symbiont is of another genus.

There are a few papers that also show how the genetics of the host does not match with that of the genetic structure of the symbiont, please read and include this in your citations:

Baums, I.B., Devlin-Durante, M.K., and LaJeunesse, T.C. (2014). New insights into the dynamics between reef corals and their associated dinoflagellate endosymbionts from population genetic studies. *Mol. Ecol.* 23, 4203-4215.

Pettay, D.T., and LaJeunesse, T.C. (2013). Long-range dispersal and high-latitude environments influence the population structure of a "stress-tolerant" dinoflagellate endosymbiont. *PLoS One* 8, e79208.

I think that the discussion could be shortened as it is redundant in a few places and too speculative in other sections. When possible, please limit speculation to subjects where there is sufficient supporting data to warrant consideration.

Breviolum muscatinei is presently not a valid species as there is no formal characterization. So it should be written in this paper as: *Breviolum 'muscatinei'*
A thorough systematic treatment is needed for this dinoflagellate to determine if it indeed constitutes one species over the entire geographic range of the U.S. coastline or comprises two or more.

Please delete the Stat et al. 2008 paper in PNAS, cited at the end of the discussion. This is a misleading work and erroneous at all levels from the experimental design, data collection, to interpretation of results. It is a poster child of how to do bad science and get it published in PNAS if you know a member who is willing to see it though without much review (that practice has since been discontinued for reasons such as this paper). Sorry to be so blunt.

Regards
Todd C. LaJeunesse

Author's Response to Decision Letter for (RSPB-2020-2896.R0)

See Appendix A.

Decision letter (RSPB-2020-2896.R1)

10-Feb-2021

Dear Dr Cornwell

I am pleased to inform you that your manuscript entitled "Genetic structure in the endosymbiont *Breviolum 'muscatinei'* is correlated with geographic location, environment, and host species" has been accepted for publication in *Proceedings B*.

Open Access

Paper charges

Sincerely,

Appendix A

Associate Editor Board Member: 1

Comments to Author:

Dear Brendan and Luis

Your manuscript has been reviewed by two experts in the field. They both like your manuscript but highlight a number of points that should be addressed before the manuscript is ultimately published. I hope you will find their reviews useful in the final edit of this ms.

Warm Regards

Line

Reviewer(s)' Comments to Author:

Referee: 1

Comments to the Author(s)

Summary

Cornwell & Hernandez analyzed the population genetics of congeneric sea anemones and their shared endosymbiont along the west coast of North America using a RADseq approach. They find little population structure in the hosts, instead identifying an isolation-by-distance pattern. But the symbionts exhibit more structure, with at least seven populations across the biogeographic range. Genetic structure in the symbiont is influenced by macroscale (geography), mesoscale (intertidal zonation), and microscale (host specificity) processes. Given the symbiont's shorter generation time and greater structure (due to low gene flow and/or strong selection), the authors conclude that the symbionts are better positioned to adapt to climate change, which may influence the adaptive capacity of the holobiont.

This is an excellent manuscript: thoughtful, easy to read, and interesting. The methods are sound and the conclusions follow from the data. I wouldn't say the results are particularly surprising (many of the patterns have been identified previously, albeit among symbiont species, not within species), but there are few studies in this intra-species space and even fewer that take advantage of high-throughput approaches to look at population structure. This focus on within-species variation does provide novel insight into the relative importance of coevolution among partners. I was particularly impressed with the analyses of correlations between genetic and environmental variation in the symbiont, as well as covariance between host and symbiont loci. I have not seen such efforts in the Symbiodiniaceae literature before; I hope this manuscript will popularize them. The Anthopleura-Breviolum association is a fantastic system; it's fairly unique in that it allows for consideration of macro/meso/micro-scale processes simultaneously. I really don't have much to critique here—just a few suggestions for minor improvements. Great

job.

Minor Comments

You should probably include the Symbiodiniaceae revision in the introduction, as it provides context for the within/between genus discussion. LaJeunesse et al. 2018 *Current Biology* (“Systematic revision of Symbiodiniaceae highlights the antiquity and diversity of coral endosymbionts”).

>> Done l. 63-68

I found two instances the taxonomic rank should be adjusted. At L301: change “within-clade” to “within-genus.” At L447: change “within symbiont genera” to “within symbiont species” (because you are talking about variation within *B. muscatinei*; because the statement holds for intra-specific variation; and because ITS2 is a species-level marker, though it also gives you the genus at the same time).

>>Done

I found the Supplemental Methods to be quite helpful for understanding the analyses and they’re only two pages long—if space is not an issue, I would recommend adding them back in. I realize *Proceedings* has a pretty stringent page limit. The discussion is a bit long, so it might be worth trying to cut it down somewhat to make more space for the methods. You might also condense figures to free up space. Figs 1 and 2 (and possibly 3) could be combined, as could Figs 4, 5, and 6.

>> We thank the reviewer for these suggestions and have consolidated the figures (1 and 2 are now a single figure, as are 4 and 5). The additional materials and methods, as written, are ~1400 additional words. After reducing the discussion, it is ~2600 words. Based on the guidance from PRSB on predicting page counts and article length, we anticipate that this article will fall just under the maximum page limit for PRSB. However, if we are incorrect in that estimation, we would be happy to incorporate more of the SI materials and methods, but have refrained from doing so due to our uncertainty about whether or not they will fit.

When discussing the BBA A. elegantissima symbiont population whose reads did not align with the *B. minutum* genome, it is worth mentioning that this likely represents a unique species, rather than just a distinct population of *B. muscatinei*.

>>Done (l. 411-414)

For the paragraph on specific environments favoring particular symbiont strains (L413-423), consider incorporating Grupstra et al. 2017 *Coral Reefs* (“Evidence for coral range expansion accompanied by reduced diversity of Symbiodinium genotypes”), as it’s relevant.

>> We thank the reviewer for bringing this to our attention and have considered it in the Discussion (although in a different section than suggested) l. 341.

Referee: 2

Comments to the Author(s)

This manuscript is an original study tracking genetic changes in symbiotic dinoflagellates over a latitudinal gradient. Its findings and the Authors' conclusions are consistent with their data and related studies as well as biological theory. Moreover, this work sets the bar higher for future approaches in how investigators ought to study symbiotic animals and their mutualistic microbes. Their most significant finding, that the symbiont is more evolutionarily responsive to selection from external environmental pressures, is substantiated by the phylogenetic differences where most symbiont lineages have evolved recently vs. the older longevity apparent in many host lineages. This is a solid contribution and with some revision should make for a nice publication.

The Sanders and Palumbi 2011 paper provided a nice study showing many of the genetic differences in the *Breviolum* symbiont found in *Anthopleura* along the coastline from Washington to California. I would like to see more effort to reconcile the present work with this paper. From the preliminary genetic work from this earlier paper, it appears that the northern population in Washington and Oregon may be a different species than that found in southern California (and possibly a third). I think that the present work would have benefited from providing at least one phylogenetic marker (LSU?) to help provide a phylogenetic framework for a better perspective of their findings. This would also reveal the identity of the symbiont found in the most southern anemones. Clearly, that symbiont is of another genus.

>>We have added additional consideration of the results of Sanders and Palumbi in the 'Macroscale' portion of the Discussion (1.300-304). Sanders and Palumbi found evidence for genetic subdivision of the symbionts roughly on the scale of 100's of km, similar to the spatial scales over which we found similar levels of differentiation. Sanders and Palumbi 2011 only consider populations from California (Pelican Point to San Diego). We have very small sample sizes from Oregon, and while they do form a cluster in the ADMIXTURE analysis, they do not form their own subpopulation, and form a group with a central CA population, which is also consistent with the PCA. Given our small sample size from Oregon, we are refraining from making assertions about whether those individuals belong to a separate species. Unfortunately, we do not have LSU data for these individuals, and were unable to generate additional data to incorporate into this study between submissions to the journal.

There are a few papers that also show how the genetics of the host does not match with that of the genetic structure of the symbiont, please read and include this in your citations:

Baums, I.B., Devlin-Durante, M.K., and LaJeunesse, T.C. (2014). New insights into the dynamics between reef corals and their associated dinoflagellate endosymbionts from population genetic studies. *Mol. Ecol.* 23, 4203-4215.

>>Added l. 297

Pettay, D.T., and LaJeunesse, T.C. (2013). Long-range dispersal and high-latitude

environments influence the population structure of a "stress-tolerant" dinoflagellate endosymbiont. PLoS One 8, e79208.

>>Added l.320-324

I think that the discussion could be shortened as it is redundant in a few places and too speculative in other sections. When possible, please limit speculation to subjects where there is sufficient supporting data to warrant consideration.

>>We have shortened the discussion and attempted to limit speculation throughout.

Breviolum muscatinei is presently not a valid species as there is no formal characterization. So it should be written in this paper as: *Breviolum* 'muscatinei' A thorough systematic treatment is needed for this dinoflagellate to determine if it indeed constitutes one species over the entire geographic range of the U.S. coastline or comprises two or more.

>>This has been corrected throughout the text and supplemental

Please delete the Stat et al. 2008 paper in PNAS, cited at the end of the discussion. This is a misleading work and erroneous at all levels from the experimental design, data collection, to interpretation of results. It is a poster child of how to do bad science and get it published in PNAS if you know a member who is willing to see it though without much review (that practice has since been discontinued for reasons such as this paper). Sorry to be so blunt.

>> This remained in the citation list after a previous round of revisions where it was no longer cited in the text. The citation has been removed from the references and we thank the reviewer for bringing our attention to problems with the study.

Regards

Todd C. Lajeunesse